# Towards full-colour tunability of inorganic electrochromic devices using ultracompact fabry-perot nanocavities

Zhen Wang [1], Xiaoyu Wang[1], Shan Cong[1], Jian Chen[1], Hongzhao Sun[1], Zhigang Chen[1], Ge Song[1], Fengxia Geng[2], Qin Chen [3] & Zhigang Zhao[1]*

Intercalation-based inorganic materials that change their colours upon ion insertion/extraction lay an important foundation for existing electrochromic technology. However, using only such inorganic electrochromic materials, it is very difficult to achieve the utmost goal of full-colour tunability for future electrochromic technology mainly due to the absence of structural flexibility. Herein, we demonstrate an ultracompact asymmetric Fabry-Perot (F-P) nanocavity-type electrochromic device formed by using partially reflective metal tungsten as the current collector and reflector layer simultaneously; this approach enables fairly close matching of the reflections at both interfaces of the $WO_3$ thin layer in device form, inducing a strong interference. Such an interference-enhanced device that is optically manipulated at the nanoscale displays various structural colours before coloration and, further, can change to other colours including blue, red, and yellow by changing the optical indexes (n, k) of the tungsten oxide layer through ion insertion.

[1] Key Lab of Nanodevices and Applications, Suzhou Institute of Nano-Tech and Nano-Bionics, Chinese Academy of Sciences (CAS), Suzhou 215123, P. R. China. [2] College of Chemistry, Chemical Engineering and Materials Science, Soochow University, Suzhou 215123, China. [3] Institute of Nanophotonics, Jinan University, Guangzhou 511443, China. *email: zgzhao2011@sinano.ac.cn

Electrochromism, which denotes a reversible change in the electronic structure and optical properties (transmittance, reflectance, or absorption) of certain materials caused by stimulation by current or potential, has attracted intense scientific and technological interests since Deb's pioneering studies due to its potential applications in displays, smart windows, and energy conservation devices[1–7]. Typically, conventional electrochromic devices can be categorized into two types by structure, namely, transmissive-type and reflective-type. No matter what type of device structure is adopted, the colours of such electrochromic devices usually rely on the intrinsic material properties. Inorganic electrochromic materials are expected to have some significant advantages over organic molecules or polymeric materials, such as high cycling, good thermal and chemical stability, and long durability, thus are at the forefront of commercialization[8,9]. However, inorganic electrochromic materials often suffer from a major weakness in intrinsic material behaviour, that is, poor colour-tuning versatility[10–12]. This can be understood from the following two points. (1) The electrochromic switching states of inorganic electrochromic materials are rather monotonous. As observed for the most widely investigated inorganic electrochromic material, tungsten oxide only depicts a onefold colour modulation from transparent to blue[13–18]. One particular example is vanadium oxide ($V_2O_5$), with the appearance in three colours (yellow ↔ green ↔ blue) during ion insertion, however, still plagued with problems of poor colour-tuning versatility, low colour saturation, and low coloration efficiency[19,20]. (2) Inorganic electrochromic materials lack the ability for subtle colour tuning[21]. For example, blue is one of the primary colours that exists in the cool spectrum. It should be noted that there is a rich variety of blue shades such as sky blue, aqua blue, ocean blue, turquoise blue, peacock blue, navy blue, etc., depending on the variation of hues, chromas and lightnesses. But for common electrochromic devices based on tungsten oxide materials, it seems that only a single blue colour with altered lightness can be obtained under different applied potentials. So far, a few attempts have been made to construct structural colour-enhanced electrochromic devices, including the introduction of opal/inverse opal photonic crystals[22–24], Bragg mirrors[25,26] and Mie scattering[27] into electrochromic devices. These modifications have attempted to introduce the capacity to display additional colours by manipulating the photonic bandgaps of the periodic structures of electrochromic materials. However, to the best of our knowledge, rich and subtle colour adjustment has rarely been achieved by inorganic electrochromic materials incorporating these modifications, and this limitation has already become a bottleneck for the further development of electrochromic technology. Given this bottleneck, it is crucial to develop novel structures for inorganic electrochromic devices to broaden their colour palettes.

A Fabry-Perot cavity is an optical resonator typically made from two facing parallel mirrors in which a light field can be selectively enhanced through resonance. Being simple and compact, these cavities have been frequently used in the design of certain optical devices, including tuneable optical filters, modulators, and pressure sensors[28–31]. Aside from very limited examples, Fabry-Perot type cavities have rarely been incorporated into electrochromic devices, especially when those devices require complex light modulating capabilities[32,33].

Herein, we propose and demonstrate an ultracompact asymmetric F–P nanocavity-type electrochromic device by replacing typical current collectors with a partially reflective metallic tungsten (W) layer, which enables fairly close matching of the reflections at both interfaces of the tungsten oxide thin layer in such a modified electrochromic device. As a result, distinct nanocavity resonances are realized in the reflection spectra, with a large spectral intensity contrast for vivid colouration compared to the flat spectrum of conventional reflective-type electrochromic devices. Accordingly, for this type of electrochromic device, a wide range of brilliant and highly saturated structural colours can be generated prior to applying voltages, with even more subtle chromatic states being achieved during the electrochromic process. Exceptionally, a rich variety of blue shades, such as sky blue, aqua blue, ocean blue, turquoise blue, peacock blue, and navy blue, can be obtained as a result of the special structural design.

## Results

**Standard electrochromic-process compatible manufacturing.** Typically, electrochromic devices consist of an electrochromic layer as a working electrode and a counter electrode with an electrolyte layer placed between to separate the two electrodes, with an electrochromic layer such as a $WO_3$ layer at the core. For conventional transmissive-type electrochromic electrodes using transparent conductive substrates such as fluorine-doped tin dioxide (FTO) glass as the current collector (Fig. 1a), simulation results suggest that the intensities of the light transmitted are almost flat and near unity over a wide spectral range from 380 nm to 800 nm, with only very small fluctuations (Fig. 1b). As a consequence, there is little perceptible structural colour in transmissive-type electrochromic electrodes (Fig. 1c). For conventional reflective-type electrochromic electrodes using Au as the current collector (Fig. 1d), the reflectivity also stays close to 1 in the long-wavelength range of 500–800 nm, although there are some small interference peaks and valleys over the short-wavelength range from 380 to 500 nm (Fig. 1e). Therefore, conventional reflective-type electrochromic electrodes usually only exhibit inconspicuous structural colours within a restricted yellow-relevant colour range, regardless of thickness, which is easily confused with the colour of Au and ignored by researchers (Fig. 1f). In contrast, when a metallic W layer serves as both a reflector and a current collector in electrochromic electrodes (Fig. 1g), distinct resonances appear in the reflection spectra, with a maximum peak-to-valley fluctuation as large as 56% (Fig. 1h). As a result, light of some wavelengths is selectively reflected, while light of other wavelengths passes through or is absorbed through a property called optical interference. Therefore, a galaxy of brilliant structural colours is expected to appear in the modified electrochromic electrodes due to the observed wavelength-selective reflections, as changes in the thickness of the $WO_3$ layer (Fig. 1i). The special structural colours could be further enriched by combination with electrochromic colours during the electrochromic process. In fact, such a simple structure just constitutes a type of unconventional asymmetric Fabry-Perot (F–P) nanocavity, wherein light is reflected back and forth at both the top and bottom surfaces of the $WO_3$ thin layer, enhancing or suppressing the light transmission/reflection to generate various structural colours, but to the best of our knowledge, this strategy has never been used in multicolored electrochromic devices.

To demonstrate the effectiveness of our design, a unique electrochromic working electrode is fabricated by successively depositing ultrathin tungsten metal and tungsten oxide films (thicknesses of ~100 and 150–250 nm, respectively) on polyethylene terephthalate (PET) or glass substrates without fluorine-doped tin oxide (FTO) or indium-tin-oxide (ITO) in a DC-magnetron sputtering system, in which only a single tungsten target is used in different sputtering atmospheres (see details in Supplementary Table 1). Note that such a process is largely compatible with the existing commercial standard electrochromic process, which is certainly an advantage for our devices. The good

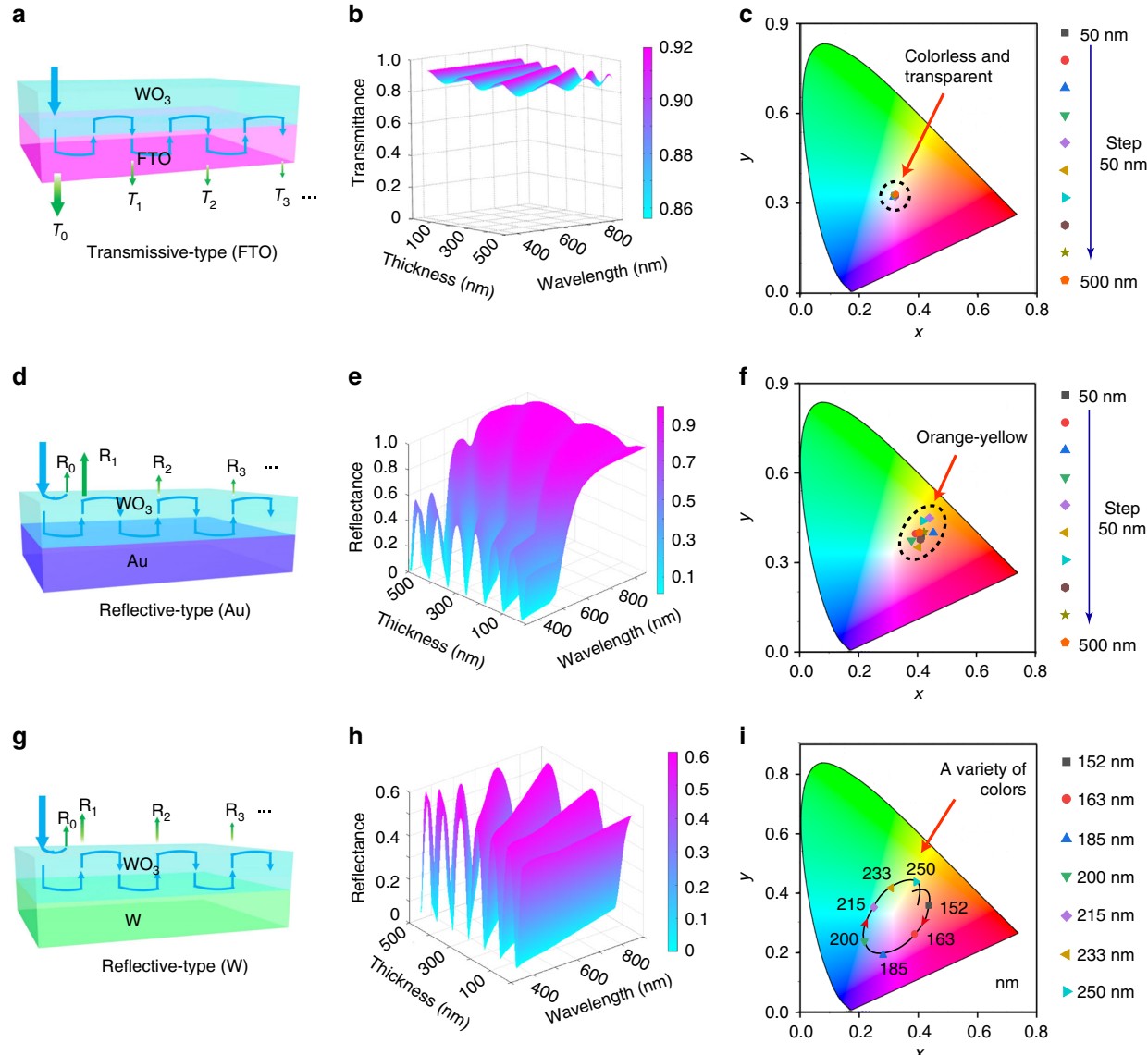

**Fig. 1 Prospective comparison of structural colours in three types of electrochromic electrodes. a, d, g** Schematic diagrams of light propagation in three types of electrochromic electrodes: electrochromic electrodes using FTO, Au, and metallic W layers as the current collectors. Simulated transmittance/reflectance spectra (**b, e, h**) and calculated colour coordinates (**c, f, i**) for the three types of electrochromic electrodes. Source data are provided as a Source Data file.

uniformity of the sputtered tungsten metal and tungsten oxide films has been confirmed by scanning electron microscope (SEM) and atomic force microscope (AFM) inspections of their surface and cross-section morphologies. The cross-section SEM image of the electrode in Fig. 2a shows a perfect bilayer assemblage consisting of a tungsten metal thin layer of 100 nm and a tungsten oxide thin layer of 200 nm, with very small thickness variation. The corresponding EDX mapping also indicates that a uniform distribution has been achieved for the tungsten layer and tungsten oxide layer in the F–P nanocavity (Fig. 2b, c). The top surface of the electrode is also seen to be rather homogenous, smooth, and dense, without any voids by analysis of AFM images (Fig. 2d). Detailed analyses of the XRD pattern of the sputtered W layer show that broadened peaks of β-W (JCPDS Cards 47-1319) are observed for the metallic W layer, while the XRD pattern of the tungsten oxide layer does not show any characteristic peaks except an amorphous shoulder, indicating that the tungsten oxide layer is amorphous (Fig. 2e). Accordingly, the equations describing the reflective behaviors of light incident in such an

electrochromic electrode can be given by Eqs. (1–4):

$$R = |\tilde{r}|^2 = \frac{\tilde{r}_{01}^2 + \tilde{r}_{12}^2 + 2\tilde{r}_{01}\tilde{r}_{12}\cos 2\tilde{\beta}}{1 + \tilde{r}_{01}^2\tilde{r}_{12}^2 + 2\tilde{r}_{01}\tilde{r}_{12}\cos 2\tilde{\beta}} \quad (1)$$

$$\tilde{r} = \frac{\tilde{r}_{01} + \tilde{r}_{12}e^{2i\tilde{\beta}}}{1 + \tilde{r}_{01}\tilde{r}_{12}e^{2i\tilde{\beta}}} \quad (2)$$

$$\tilde{r}_{01} = (\tilde{n}_0\cos\tilde{\theta}_0 - \tilde{n}_1\cos\tilde{\theta}_1)/(\tilde{n}_0\cos\tilde{\theta}_0 + \tilde{n}_1\cos\tilde{\theta}_1) \quad (3)$$

$$\tilde{r}_{12} = (\tilde{n}_1\cos\tilde{\theta}_1 - \tilde{n}_2\cos\tilde{\theta}_2)/(\tilde{n}_1\cos\tilde{\theta}_1 + \tilde{n}_2\cos\tilde{\theta}_2) \quad (4)$$

where R is the total reflectivity in such a structure, $\tilde{r}_{01}$ and $\tilde{r}_{12}$ are the reflection coefficients at the air/WO$_3$ and WO$_3$/W interfaces, $\tilde{\beta} = (2\pi/\lambda)\tilde{n}_1 d\cos\tilde{\theta}_1$ is the phase thickness of the WO$_3$ layer, $\tilde{n}_j$ ($j = 0,1,2$) is the complex refractive index of air, the WO$_3$ layer and the W layer, respectively, $\tilde{\theta}_j$ ($j = 0,1,2$) is the incident angle of each layer, and $\lambda$ is the wavelength. By Eqs. (1–4), one can see

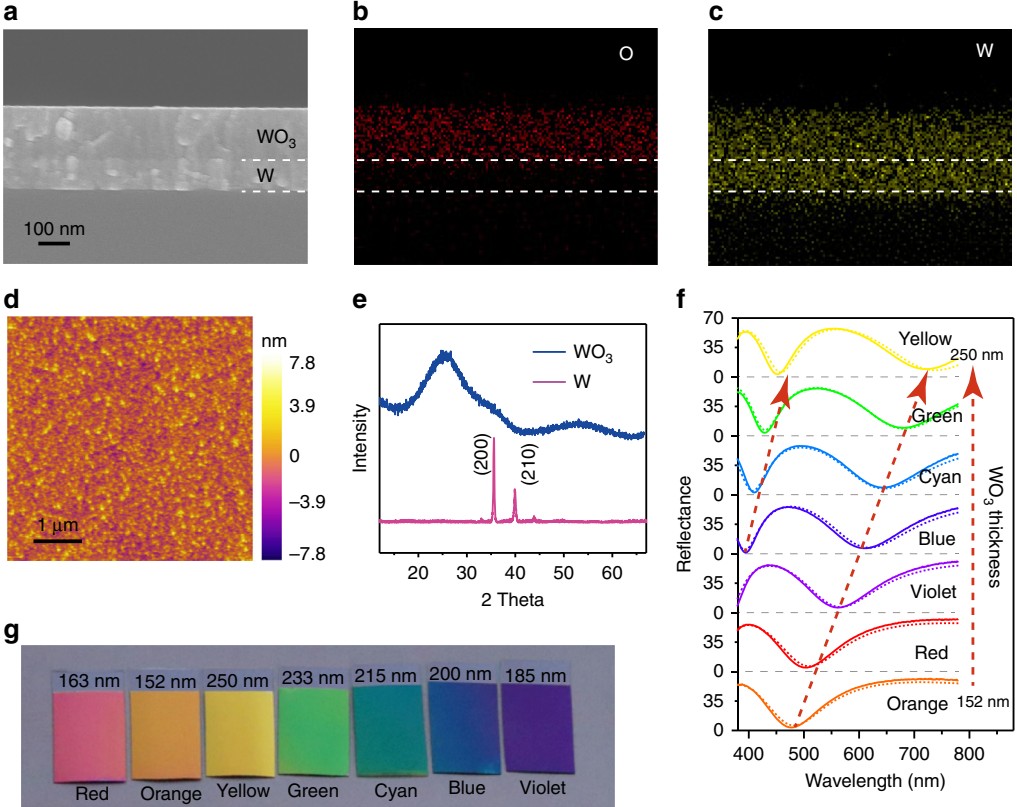

**Fig. 2 Structural characterizations of our F–P nanocavity-type electrochromic electrodes. a–c** Cross-sectional SEM image and corresponding elemental mapping images of O and W. **d** AFM image. **e** XRD pattern. **f** Simulated (dashed lines) and measured (solid lines) reflection spectra with different thickness of $WO_3$ increasing from 152 nm to 163, 185, 200, 215, 233, and 250 nm. **g** Optical images of the electrochromic electrodes with different thicknesses of the $WO_3$ layer. Source data are provided as a Source Data file.

that the $WO_3$ film thickness and the refractive indexes of the W film and the $WO_3$ film are three major factors contributing to the modulation of the spectral reflectance at the two interfaces.

Finite difference time domain (FDTD) calculations render clear understanding of the relationship between the resonance behaviour and the $WO_3$ thickness. Figure 2f shows the calculated reflection spectra of a modified electrochromic electrode with varied $WO_3$ thicknesses of 152, 163, 185, 200, 215, 233, and 250 nm, respectively, when the thickness of the underlying metallic W layer is fixed to be 100 nm. For the electrode with a 152 nm $WO_3$ thickness, there is a broad reflectance valley near 477 nm in the calculated reflection spectrum. Upon increasing the thickness of the $WO_3$ layer to 163, 185, 200, 215, 233, and 250 nm, the location of the reflection valley is largely redshifted from 477 to 700 nm, while a new reflection valley appears in the short-wavelength region (400–450 nm). As in the case of the electrode with a 250 nm $WO_3$ thickness, the calculated reflection spectrum removes the significant broad valley but demonstrates two additional narrow valleys. It can be seen that the overall shape of each experimental spectrum matches quite well with the calculated spectrum at the same $WO_3$ thickness, showing similar trends in reflectance as a function of the wavelength (Fig. 2f). Accordingly, seven types of colourful colours that are quite energetic, which rarely appear in inorganic electrochromic systems, are experimentally obtained in the electrochromic working electrode with the help of the F–P nanocavity when different thicknesses of the $WO_3$ layer are used (Fig. 2g). As a matter of fact, the structural iridescent tuned colours in our modified electrochromic electrode can almost span the entire visible spectrum. By contrast, comparative studies indicate that the thickness of the metal W layer has no impact on the structural

colours of the electrochromic electrodes in the thickness range examined here (Supplementary Fig. 4). With additional measurements performed by a variable-angle spectroscopic ellipsometer, the incident angle dependence of such structures can also be determined. With additional measurements performed by a variable-angle spectroscopic ellipsometer, the incident angle dependence of such structures can also be determined. Clearly, the angle-resolved reflection spectra of our F–P nanocavity-type electrochromic working electrodes remain almost invariant at oblique angles of incidence ranging from 0° to 40° (Supplementary Fig. 4), and the corresponding angle-resolved optical images recorded at eight different angles also show no dramatic colour change from 0° to 40° (Supplementary Fig. 5). These results confirm that our F–P nanocavity-type electrochromic working electrodes are almost insensitive over a relatively wide range of incident angles up to ±40°, allowing angle-insensitive electrochromic performances.

**Wide color gamut**. Remarkably, the full-colour F–P nanocavity-type electrochromic electrode enables a unique electrochromic optical switching functionality that is quite different from that of conventional electrochromic electrodes. The colouring process and the resultant optical changes in our F–P nanocavity-type electrochromic electrode can be illustrated in Fig. 3 and are compared with those of transmissive-type $WO_3$/FTO and conventional reflective-type $WO_3$/Au electrochromic electrodes. For conventional transmissive-type $WO_3$/FTO electrochromic devices, applying negative voltages produces an overall decrease of the transmittance of the electrode without appreciably shifting the peak positions in the transmittance spectra (Fig. 3a). The

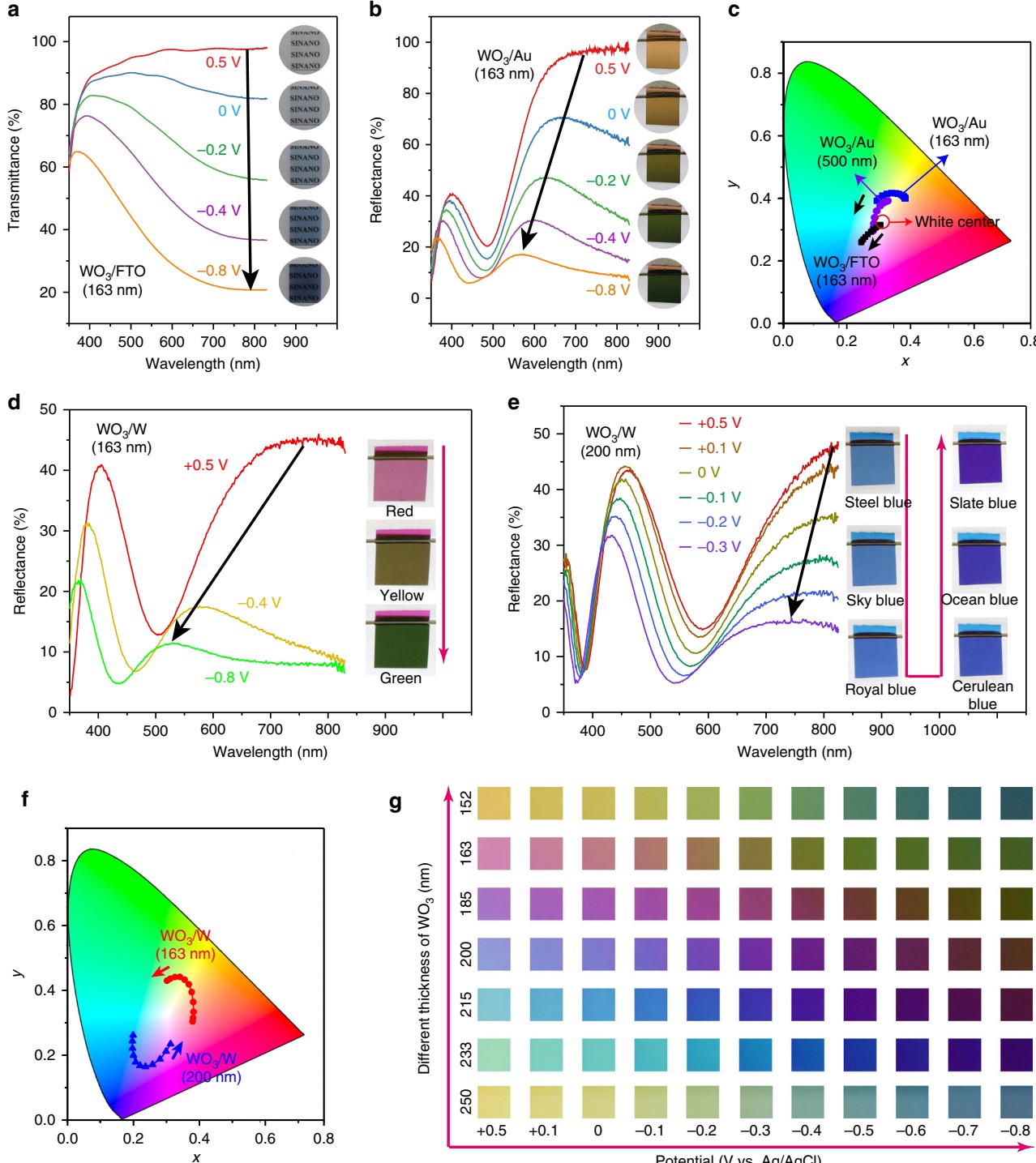

**Fig. 3 Demonstration of rich and subtle colour modulation. a** Optical transmittance spectra and corresponding optical images of a conventional transmissive-type WO₃/FTO electrochromic electrode at different applied potentials. **b** Optical reflection spectra and corresponding optical images of a conventional reflective-type WO₃/Au electrochromic electrode at different applied potentials. **c** CIE colour coordinates of a conventional transmissive-type WO₃/FTO electrochromic electrode and two typical conventional reflective-type WO₃/Au electrochromic electrodes at different applied potentials. **d–f** Optical transmittance spectra, corresponding optical images and CIE colour coordinates of our F–P nanocavity-type electrochromic electrode at different applied potentials, with WO₃ layers of 163 nm (**d**) and 200 nm thickness (**e**), respectively. (**g**) Colour gallery obtained from our F–P nanocavity-type electrochromic electrode at different applied potentials. Source data are provided as a Source Data file.

colour-rendering characteristics of conventional electrochromic electrodes during an electrochromic process can be described by its location within a CIELAB colour space (CIE 1931). It can be seen that the colour points in the colour diagram map out a straight line through the white centre (Fig. 3c), which is called an

iso-hue line that specifies colours with the same hue. Such results clearly suggest that conventional transmissive-type WO₃/FTO electrochromic devices mostly manipulate the darkness or lightness of electrochromic colours but are not competent at controlling colour tints or hues. For conventional reflective-type

WO$_3$/Au electrochromic devices, the structural colours obtained at different WO$_3$ thicknesses that can be displayed before electrochromic coloration are limited to within the small circle in the CIE diagram, confirming a single yellow-relevant colour (Au layer appears yellow) (Fig. 1f). Upon colouration, the CIE coordinates of WO$_3$/Au electrochromic devices only promise a yellow–green or dark green colour but do not include other distinct colours, which can also be clearly observed in the reflection spectra and optical images (Fig. 3b). In addition, the range of the colour span is narrow, leading to quite limited colour modulation (Fig. 3c). In great contrast, rather rich, subtle colour modulation could be achieved in our F–P nanocavity-type electrochromic electrodes. For example, in the bleached state under an applied voltage of +0.5 V, the 163-nm-thick electrode gives a reddish colour, while the colour changes to yellow and green in the coloured states under the voltages of −0.4 V and −0.8 V, respectively (Fig. 3d). Accordingly, its reflection spectrum displays a drastic hypsochromic shift from 760 to 517 nm in the peak position (Fig. 3d and Supplementary Fig. 7), thus achieving a very large modulation range (243 nm) compared with the values previously reported for inorganic electrochromic materials (for example, 48 nm for WO$_3$/NiO Bragg mirrors[25] and 72 nm for WO$_3$ Bragg mirrors[26]). Such vivid colour changes also cause a remarkable shift of the CIE colour coordinates, forming a circular area from red to green and suggesting different tints (Fig. 3f). As another example, blue colours with different colour hues can be obtained from the 200-nm-thick electrode, depending on its applied voltage. The electrode can vary from steel blue to sky blue, royal blue, cerulean blue, ocean blue and, finally, slate blue as the applied potential is scanned between +0.5 V, +0.1 V, 0 V, −0.1 V, −0.2 V, and −0.3 V (Fig. 3e). Correspondingly, the colour points on the colour diagram map out an arc shape but not an iso-hue line, demonstrating the capability for changing the colour hue (Fig. 3f). Furthermore, the CIE chromaticity coordinates of our F–P nanocavity-type electrochromic electrodes are closer to the edges of the chromaticity diagram than those of conventional electrochromic films, indicating higher-saturation colours. Through the elaborated design, the F–P nanocavity-type electrodes have been shown to be synthetically tuneable to produce a large family of multicolour states, consequently generating a near full-colour palette that has long been desired for inorganic electrochromic materials (Fig. 3g, Supplementary Figs. 7-9). The observed differences between conventional and F–P nanocavity-type electrochromic electrodes suggest that they might exert their colouring effects through different mechanisms. In conventional WO$_3$ electrochromic electrodes, colours are mainly determined by the energy band structures of the WO$_3$ layer before and after lithium intercalation. However, for our nanocavity-type electrochromic electrode, its colour changes are directly related to the large change in the refractive index of the WO$_3$ layer. The refractive index of the WO$_3$ layer is found to vary with the amount of lithium inserted, e.g. decreasing from 2.15 to 1.61 at the wavelength of 600 nm under an applied voltage of −0.8 V (Supplementary Fig. 10). As a result, the effective optical thickness is reduced, and the resonant reflection peaks are blue-shifted, thus producing different colour appearances.

In addition to colour modulation, other important electrochromic parameters such as coloration efficiency, switching time, and cycling stability have also been evaluated for our F–P nanocavity-type electrochromic electrode. An interesting feature of our electrochromic electrode is that the metal W layer is used not only for the construction of the F–P nanocavity but also as a current collector in the electrochromic system. Compared to electrochromic electrodes incorporating traditional current collectors such as FTO and ITO, the metal W-covered F–P nanocavity-type electrochromic electrode maintains a comparable

electrochemical performance under the same experimental conditions, which is evidenced by the almost similar cyclic voltammograms at the example scan rate of 10 mV s$^{-1}$ (Supplementary Fig. 11). The colouration efficiency (CE) is a very important parameter for electrochromic materials, which is measured by monitoring the amount of injected/ejected charge as a function of the change in optical density (calculation details are depicted in the Supplementary Information). Usually, a high value of CE indicates that the electrochromic material exhibits a large optical modulation upon insertion (or extraction) of a small charge amount. Supplementary Figure 12 plots the relation between optical density and current density, and the CEs for electrochromic electrodes can be calculated from the slope of the linear region of these curves. As shown in Supplementary Fig. 12, the CEs of traditional FTO- and ITO-covered electrochromic electrodes are calculated to be 44.4 cm$^2$ C$^{-1}$ and 47.1 cm$^2$ C$^{-1}$, respectively. The F–P nanocavity exhibits a CE of 48.6 cm$^2$·C$^{-1}$, which is slightly higher than those of the normal current collector-covered electrochromic electrodes. The switching time between colouration and bleaching in electrochromism is another important parameter in the electrochemical process, which is defined as the spanning period required for 90% of the change between the steady bleached and coloured states (Supplementary Fig. 13, Supplementary Table 2). For traditional commercial FTO- and ITO-covered electrochromic electrodes, the coloration times are found to be 2.9 and 7.5 s, and the bleaching times are 4.2 and 8.7 s, respectively. For the metal W (100 nm thickness)-covered F–P nanocavity-type electrochromic electrode, the colouration time is 3.3 s, and the bleaching time is 3.1 s. Obviously, the switching speed of the metal W-covered F–P nanocavity-type electrochromic electrode is faster than that of the ITO-covered electrochromic electrode but is comparable to that of the FTO-covered electrochromic electrode. In addition, the F–P nanocavity-type electrochromic electrode also exhibits good cycling stability, retaining 81% of its initial charge capacity after 1000 cycles (Supplementary Fig. 14).

**More elaborate device designs**. In the above results, note that the colour modulation is achieved by using a pair of asymmetric mirrors (tungsten/tungsten oxide/air) in the F–P nanocavity to yield wavelength-dependent reflectance. It is interesting to further explore the performances of F–P nanocavity-type electrodes when other mirrors are adopted, for example, upon adding a top metal mirror (Fig. 4a). In such cases, the equations describing the total reflectivity of our F–P nanocavity-type electrochromic electrodes are modified as Eqs. (5) and (6):

$$R = |\tilde{r}|^2 = \left| \frac{\tilde{r}_{0m} + r'e^{2i\tilde{k}_m}}{1 + \tilde{r}_{0m}r'e^{2i\tilde{k}_m}} \right|^2, \text{ where } r' = \frac{\tilde{r}_{m1} + \tilde{r}_{12}e^{2i\tilde{\beta}}}{1 + \tilde{r}_{m1}\tilde{r}_{12}e^{2i\tilde{\beta}}} \quad (5)$$

$$\tilde{r}_{0m} = (\tilde{n}_0 \cos \tilde{\theta}_0 - \tilde{n}_m \cos \tilde{\theta}_m)/(\tilde{n}_0 \cos \tilde{\theta}_0 + \tilde{n}_m \cos \tilde{\theta}_m) \quad (6)$$

where $\tilde{r}_{0m}$ and $\tilde{r}_{m1}$ are the reflection coefficients at the air/metal and metal/WO$_3$ interfaces, $\tilde{\beta}_m = (2\pi/\lambda)\tilde{n}_m h \cos \tilde{\theta}_m$ is the phase thickness of the top metal layer, and $\tilde{n}_m$, $h$, and $\tilde{\theta}_m$ are the complex refractive index, thickness, and incident angle of the top metal, respectively. The most remarkable aspect of this modification achieved by adding a top metal mirror is relevant to the flexible control and substantial variation of the colour saturation of our F–P nanocavity-type electrochromic electrodes. For example, two representative metals, Ag with a low refractive index and high extinction coefficient and W with a high refractive index and high extinction coefficient, are selected to investigate the effects of the top metal layer on colour saturation. In the case of the Ag top layer, the resonance dips in the reflection spectra of

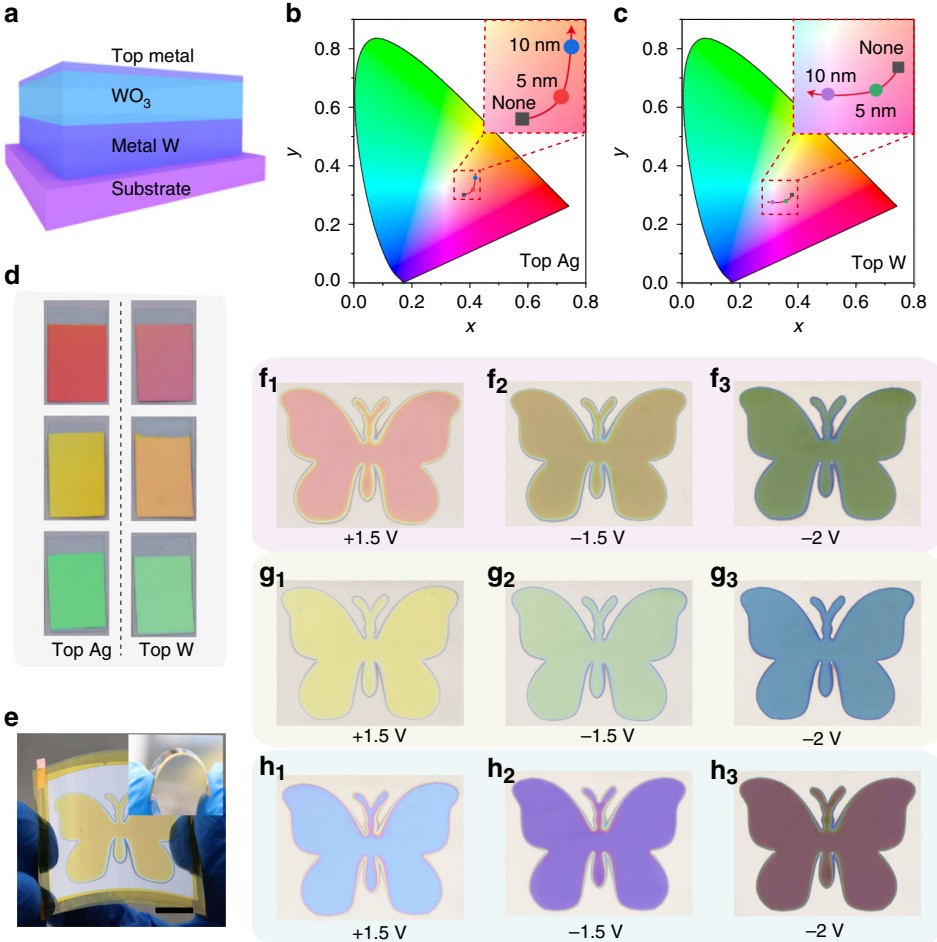

**Fig. 4 Optimization design and prototype demonstration. a** Schematic illustration of the proposed F–P nanocavity-type electrochromic electrodes with a top metal layer. **b, c** CIE colour coordinates for F–P nanocavity-type electrochromic electrodes with Ag and W top layers. **d** Typical optical images of the electrochromic electrodes with Ag and W top layers. **e** Schematic sketch of the electrochromic device prototype built from our colourful F–P nanocavity-type electrochromic electrodes. Scale bar: 1 cm. Inset: extreme bending flexibility demonstrated by wrapping an electrochromic device around sharp 90-degree corner. **f–h** Different colour states (applied voltages of 1.5, –1.5, and –2 V, respectively) of three butterfly-like electrochromic devices containing the three basic colours of red, yellow, and blue, where the thickness of tungsten oxide layer is carefully chosen to be 163, 152, and 200 nm, respectively. Source data are provided as a Source Data file.

these electrochromic electrodes become greater (Supplementary Figs. 15, 16), and the corresponding coordinate points move obviously towards the edges of the chromaticity diagram (Fig. 4b), suggesting evidence for higher colour saturation compared to that of the electrochromic electrodes without a top coat. In contrast, the W top layer plays an opposite role in controlling the colour saturation, resulting in a noticeable decrease in colour saturation (Fig. 4c). Typical optical images of the electrochromic electrodes with and without a top layer also demonstrate the changes in colour saturation: the electrochromic electrode with an Ag top layer looks bright, while the electrode with a W top layer is rather dim and dark (Fig. 4d).

Finally, our colourful F–P nanocavity-type electrochromic electrode has been integrated into a flexible prototype electrochromic device with complex butterfly patterns. Specifically, Favrini films (a type of photographic film comprising of PC/PP/PET/PVC multilayers) are used as deposition masks for patterning the electrodes on the metal W thin layer, and $WO_3$ films of different thicknesses are then sputtered onto the uncovered regions of the metal W layer to form the patterns with the desired colours, following by the assembly of prototype electrochromic devices (see details in the Supplementary Information). As demonstrated in Fig. 4e, the as-fabricated fully

packaged electrochromic devices are highly flexible and can be bent without affecting the structural integrity of the devices. Figure 4f–h show different colour states (applied voltages 1.5, −1.5, and −2 V, respectively) of three electrochromic devices with three basic colours, red, yellow, and blue, achieved through controlling the thickness of the $WO_3$ layer to be 163, 152, and 200 nm. Our butterflies can reversibly flash brilliant colours from their wings. When these devices are biased to −1.5 V, the butterflies are coloured with yellow, green, and violet colours in the three devices, respectively. Upon further increasing the voltage to −2 V, green, blue and dark red colours can be found in the three devices, respectively. To the best of our knowledge, this is the first time that so many vibrant colours have been achieved in inorganic materials-based electrochromic devices.

## Discussion

In conclusion, it has been demonstrated that a type of electrochromic device utilizing ultracompact asymmetric F–P nanocavity as the electrochromic active layer can be fabricated. The devices are structurally very simple, and fully compatible with the existing commercial standard electrochromic process, but they can display rich colour tunability with very wide colour gamut

distribution that are rather distinct from those of conventional inorganic material-based electrochromic devices. Our present method may represent an exciting development in electrochromic devices with full-colour tunability, which has long been desired for commercial electrochromic applications.

## Methods

**Preparation of magnetron sputtered films.** All films were fabricated by magnetron sputtering, with the preparation details summarized in Supplementary Table 1. Favrini film masks were purchased from Jili Company, China.

**Fabrication of electrochromic display devices.** Electrochromic display devices were assembled using the F–P nanocavity film as the electrochromic layer, an NiO film as the ion storage layer, and 1 M lithium perchlorate ($LiClO_4$) and poly(methyl methacrylate) (PMMA) in propylene carbonate (PC) as the electrolyte, respectively. The patterned F–P nanocavity electrode was prepared on a PET substrate, and an NiO film as the counter electrode was placed onto the ITO/PET substrate. Then, two slim copper conductive adhesives were attached onto the conductive surface of the F–P nanocavity electrode and the counter electrode, respectively. 1-mm-thick spacer were glued on the four edges of patterned F–P nanocavity electrode with a small free space on the top edge. Then the counter electrode was pressed onto the patterned F–P nanocavity electrode in such a way that the two coatings faced each other, followed by filling with 1 M $LiClO_4$/PC (propylene carbonate) with 8 wt % PMMA through syringe injection to complete the device structure. The devices were finally sealed using adhesive epoxy.

**Electrochemical and optical measurement.** The reflection spectra were collected by a UV–vis spectrophotometer (V660, JASCO), and the angle-resolved reflection spectra were collected by an angle-resolved spectrum system (R1, ideaoptics, China) equipped with spectrometers (NOVA-EX, ideaoptics, China) over a wavelength range of 350–850 nm. The electrochemical behaviours were studied by means of cyclic voltammetry (CV) and chronoamperometry (CA), both performed on an electrochemical workstation (CHI 760, CH Instruments, Inc., China) using a conventional three-electrode test cell for the single F–P nanocavity electrochromic electrode. A platinum wire and Ag/AgCl saturated with KCl were used as the counter and reference electrode, respectively. A two-electrode system was used for the performance testing of the fabricated electrochromic devices at room temperature.

**Optical constants of W and $WO_3$.** Refractive index (n) and extinction coefficient (k) were probed by spectroscopic ellipsometry measurements for single-layer films deposited on bare silicon substrates under different incident angles, with the data analysis performed using WVASE32 software (Supplementary Figs. 1, 2).

## Data availability

The data that support the findings of this study are available from the corresponding authors on reasonable request. Additionally, data reported herein have been deposited in the Figshare database, and are accessible through https://doi.org/10.6084/m9.figshare.11154791.

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

## Acknowledgements

This work was supported by the National Natural Science Foundation of China (51572286, 51772319, 51772320, and 51972331). Z.G.Z. would like to acknowledge the support from the Natural Science Foundation of Jiangxi province (20181ACB20011), the External Cooperation Program of the Chinese Academy of Sciences (121E32KYSB20190008), Six Talent Peaks Project of Jiangsu Province (XCL-170), and the Science and Technology Project of Nanchang (2017-SJSYS-008). S.C. would like to acknowledge the support from the Youth Innovation Promotion Association, CAS (2018356), Outstanding Youth Fund of Jiangxi Province (20192BCBL23027), and the Suzhou Industrial Science and Technology Program (SYG201737). Q.C. would like to acknowledge the support from the National Key Research and Development Program of China (2019YFB2203402).

## Author contributions

Z.Z. conceived the project. Z.W., Z.Z., F.G. and Q.C. designed the experiments and analyzed the data. Z.W., J.C., H.S., Z.C. and G.S. performed material synthesis, structural characterization, devices fabrication, and electrochromic measurements. X.W. contributed with the parts of simulation and calculation. Z.W., S.C. and Z.Z. co-wrote the paper. All authors discussed the results and commented on the manuscript.

## Competing interests

The authors declare no competing interests.
