## [Peer Review File · Nature Communications]

Reviewers' comments:

Reviewer #1 (Remarks to the Author):

The manuscript "Towards Full-colour Tunability of Inorganic Electrochromic Devices Using Ultracompact Fabry-Perot Nanocavities" presents very exciting results that without question are of great interest in the electrochromic community. I think this work is very important to the whole electrochromic community, and thus I recommend publishing this manuscript. The specific comments are as follows:

1. When I am reading this manuscript, especially the section of Figure 1 in the manuscript, I have a question regarding the choice of the metal layer. Why the authors choose the W as the metal layer? Will the Al or Ag layer also work for the multi-color electrochromic devices? This is just for my curiosity.
2. In the demonstration of the flexible device part, I have a concern that will the NiO counter layer block the color originated from the WO₃ layer?
3. The paper can be written in a more concise fashion. For example, it seems that the "capacity" in line 234 should be "capability". I am also confused about what is "Favrini films" in the line of 334. In the line of 338, the assembly details of the flexible device are shown in the methods part rather than in the SI part.

Reviewer #2 (Remarks to the Author):

In this manuscript, the authors have developed a novel Fabry-Perot nanocavity-type electrochromic device with rich and subtle color modulation, which is a breakthrough in electrochromic filed, especially for inorganic materials. Based on the structural colors generated among layered W and WO₃ interfaces, an all-inorganic electrochromic device has been demonstrated with nearly full-color tunable range, providing a new strategy for the achievement of multichromatic electrochromism with inorganic materials. Since the design of device is conceptually novel together with interesting findings, this manuscript is suggested to be suitable for publishing in Nature Communications after minor revisions.

- (1) In page 4 line 53, the authors claimed "Inorganic electrochromic materials lack the ability for subtle colour tuning". This statement is not rigorous because V₂O₅ is one multicolor inorganic electrochromic material.
- (2) In page 4 and page 5, the authors described that the Figure 1 is the data of electrochromic device. However, it can be seen from Figure 1a, d and g, all the data should be the electrochromic electrode, not the electrochromic device.
- (3) In Figure 2f, the reflectance valley of the F-P nanocavity-type electrode is observed to be red-shifted as the thickness of WO₃ increased. Could the authors explain for it?
- (4) In Figure S8, the specific values of applied voltages and thicknesses of WO₃ should be provided.
- (5) I am curious weather the reflective metallic tungsten layer is unique in the Fabry-Perot nanocavity-type electrochromic device. Have the authors made a comparison between tungsten and other metals as reflective layers, for the performance and mechanism in the relative Fabry-Perot nanocavity-type electrochromic devices?
- (6) The references in this manuscript are few. There are some latest works focus on combination of structural and electrochromic effects (Angew. Chem. Int. Ed. 2016, 55, 2503 –2506); amorphous WO₃ with ultra-large optical modulation (Chem. Sci., 2016, 7, 1373–1382); WO₃/metal tungsten electrochromic electrode (Solar Energy Mater. Solar Cells, 2011, 95, 2107); novel chromic mechanism based on light scattering theory (ACS Appl. Mater. Interfaces 2018, 10, 37685); which are highly related to the present work, recommended to the authors for discussion.

Reviewer #3 (Remarks to the Author):

The authors present an interference-based approach to obtain distinct colors that may be further transitioned between a palette of available colors based on the electrochromic characteristics of the materials used.

More precisely, color tuning is achieved by means of thickness control, while further color transitions are obtained via electrochemical control. Tungsten oxide and metallic tungsten are used as the base materials to obtain the interference patterns, in the architecture that authors describe as Ultracompact Fabry-Perot nanocavities.

Overall, the manuscript presents a sound study that may impact the electrochromic field, opening new directions for broader color tunability in inorganic materials.

Acceptance for publication in Nature Communications may be considered, providing several changes are properly addressed.

Please let me suggest some of these changes/comments (not in order of importance, just randomly listed as they appear while reading the text):

1. Please avoid the use of subjective adjectives or descriptions like "dexterously" (line 24 and 66) or "gorgeous" (line 169), in favor of a more aseptic and descriptive style.
2. Lines 27-32 are directly copied from the main text (lines 71-75). Please elaborate a different sentence.
3. A more detailed description related to the results shown in figure 1 is missing. There is no information on how the simulated data were obtained. Please include a detailed explanation (whether in main text or Supp. Info).
4. While reading the manuscript, one miss additional efforts on trying to teach the working principles and fundamentals of this architecture. Specially, it would be important to teach why color tuning works with one material as a reflector (W) but not with others (Au). Although it may be obvious for some specialists in the field, it may not be such for all people in the electrochromics community. This would also be particularly important and encouraged considering the broad and multidisciplinary audience of this journal.
5. As commented before for the simulations, information about the FDTD calculations should be included (in supp. Info). (lines 154-155) Related to those lines, there could be a mistake in the text. I don't see that the text corresponds to the refereed figures S1 and S2. Dependence of the resonance behavior vs thickness is not shown in those figures.
6. Figure 2f. Please specify which lines (dashed or solid) correspond to simulated and measured data
7. Lines 177- 182. I think refereed figures 4 and 5 are interchanged. Lines 178-180 correspond to fig 5 and the rest to fig 4?
8. Line 184. Attending to the spectra shown, it would probably be better to state a $\pm 40^\circ$ insensitivity range, instead of $\pm 50^\circ$ as the authors claim. As a related comment, I would say that this angle range probably is not outstanding, compared with other image-forming competing technologies (electronic ink or LED). From my point of view, this is one of the weaknesses of this technology as far as it is presented now. The authors may want to comment on it and propose ways of overcoming or improving these limitations.
9. Line 193- The authors may want to find another equivalent expression to "rich-colours contained electrochromism". I think is confusing.
10. One of my main concerns about the presented technology is the lack of a transparent to colored transition. While multicolor transitions may be appealing, they are difficult to use in real devices. For instance, if the authors would like to develop a full color pixel based on a monolithic architecture, they will not have the possibility of showing a non- colored state, which greatly lowers the future development of this technology, and restricts it to a limited number of applications. In the same direction, the intrinsic architecture of these devices is limited to a reflective type. This compares unfavorably with other existing electrochromic approaches in which the reflective-transmissive behavior is translated to the electrolyte. The authors may have these aspects in mind in order to try to make a broader discussion on the future of this technology.

11. Line 295- Differences in the switching speeds are not so obvious as the authors claim. One could expect that ITO and FTO based devices would not show many differences as the conductivity values on these substrates is usually similar. Again, one would expect that a metallic substrate would enhance considerably the switching speed. It would be interesting to add conductivity values of each substrate. Also, surface values of the studied devices will help on this. For small surfaces , the effect of the conductivity of the substrate would be less critical.

12. Long term cyclability. I'd suggest the authors to perform a longer study. Just 150 cycles seems clearly insufficient considering the conventional tests on the field.

13. Line 340- The authors refer to a bleached state. Please be cautious while using that word (I'd suggest not to use it in this context).As I mentioned before, there is no such a bleached state in this architecture (understood as a transparent state, as usually referred in the field). "Light colored" or equivalent wording may be used.

14. Lines 360-361. I think the verb is missing in the sentence.

15. Conclusions: Please be cautious with the use of full color tunability, as it is not the same to achieve a full color palette of different films that achieving full color transitions electrochromically.

16. Some discussion on previous literature is missing in the introduction. Some references that mention or try Fabry-Perot interference based architectures are:

<https://doi.org/10.1016/j.tsf.2013.10.030> <https://doi.org/10.1080/14786437308227562> or <https://doi.org/10.1117/12.621397>

Reviewer #1 (Remarks to the Author):

The manuscript “Towards Full-colour Tunability of Inorganic Electrochromic Devices Using Ultracompact Fabry-Perot Nanocavities” presents very exciting results that without question are of great interest in the electrochromic community. I think this work is very important to the whole electrochromic community, and thus I recommend publishing this manuscript. The specific comments are as follows:

Comment 1:

When I am reading this manuscript, especially the section of Figure 1 in the manuscript, I have a question regarding the choice of the metal layer. Why the authors choose the W as the metal layer? Will the Al or Ag layer also work for the multi-color electrochromic devices? This is just for my curiosity.

Author reply: Thank you for the valuable comments. Compared with organic electrochromism, one of the greatest weaknesses of inorganic electrochromic materials is their monotonous colour changes. Thus, the ultimate goal of full-colour tunability for future electrochromic technology has been difficult to achieve with devices based on these typical materials. Our present work aims to broaden the colour versatility of inorganic electrochromic materials by introducing ultracompact Fabry-Perot (F-P) nanocavities into relative electrochromic devices. For a proposed F-P nanocavity-type electrochromic device, the metal layer is the central component, as this layer acts as a reflecting mirror for the optical interference that allows the generation of different colours. Generally, the choice of metal layer in our work is based on the following criteria.

(1) **Large dip depth.** As is well known, the depth of the resonance dip (that is, the difference in intensity between the dip minimum and the threshold) in a reflectance spectrum is an important indicator of resonance efficiency. The greater the depth of this dip, the better the efficiency of resonance that will be achieved. Supplementary Figure 17 shows the dependence of the depth of resonance dip on refractive index (n) and extinction factor (k) for the metal layers analysed in our work (for these measurements, 200-nm-thick WO_3 is set as the dielectric layer).

The yellow region of the spectrum corresponds to a large dip depth, while the blue region indicates a small dip depth. As can be seen, the perceived (n, k) values for Al, Ag, Au and Cu are either fully or partly located in the blue region, indicating that the colour gamut was limited because a strong resonance could not be excited over the whole visible range. In contrast, the (n, k) values for Ni, Ti, Cr, V, Zn and W are all located in the yellow region, suggesting that a wide colour gamut is established by strong resonance behaviours operating across the entire visible spectrum.

(2) **Good stability.** We choose metal layers with good environmental and electrochemical stability.

(3) **Strong adhesion characteristics.** We choose metal layers that adhere well to the substrate and the electrochromic layer.

According to the above criteria, W is selected as the ideal choice for the proposed F-P nanocavity-type electrochromic device.

Supplementary Figure 17. The dependence of the depth of resonance dip on the refractive index (n) and extinction factor (k) of metal layers with different thicknesses. The parameters involved in the simulation are wavelength (400~800 nm) and WO_3 thicknesses (200 nm). The optical constant (n, k) of metal W is obtained by fitting the ellipsometry spectra using a quasi-static approach (Supplementary Figure 1, 2). The optical constants (n, k) of the other metals are all cited from the following website:

<http://refractiveindex.info> (this source does not report an optical constant for metal W in the visible wavelength region).

Going further, the simulated reflectance spectra obtained from Al and Ag metal layers are given in Supplementary Figure 18. Only weak resonances are observed in the simulated reflectance spectra, with a maximum peak-to-valley fluctuation below 20%. This indicates that the reflected colours are only weakly perceptible in the cases of Ag and Al.

Supplementary Figure 18. Simulated reflectance spectra derived from Al (a) and Ag (b) metal layers.

Comment 2:

In the demonstration of the flexible device part, I have a concern that will the NiO counter layer block the color originated from the WO_3 layer?

Author reply: Thank you for the valuable comments. The optical image and transmittance spectra derived from the magnetron-sputtered NiO film are provided in Supplementary Figure 19, 20. As can be seen, the prepared films are almost transparent, with a high average transmittance of 85% over the whole visible region, demonstrating that they do not block the colour originating in the WO_3 layer. This result is consistent with the transmission spectra measured at different applied potentials for the electrochromic device consisting of two NiO electrodes paired with a transparent LiClO_4 -based electrolyte. As shown in Supplementary Figure 20, the

transmissions of the NiO electrodes and their corresponding devices remained nearly constant at the different applied potentials, suggesting that the NiO electrodes do not block the colours of the WO₃ layer.

Supplementary Figure 19. An optical image of the NiO electrode, showing its high transmittance.

Supplementary Figure 20. The transmission spectra of the NiO electrodes with (a) the corresponding electrochromic devices at different applied potentials and (b) NiO electrodes paired with a transparent LiClO₄-based electrolyte. Significantly, during voltage application, the transmissions of the NiO electrodes and their corresponding devices remained almost constant, further suggesting that the NiO electrodes do not block the colours emitted by the WO₃ layer.

Comment 3:

The paper can be written in a more concise fashion. For example, it seems that the “capacity” in line 234 should be “capability”. I am also confused about what is “Favrini films” in the line of 334. In the line of 338, the assembly details of the flexible device are shown in the methods part rather than in the SI part.

Author reply: Thank you for the valuable comments.

- (1) The word “capacity” has been replaced by “capability” in the revised manuscript.
- (2) In the line of 334, “a type of photographic film comprising of PC/PP/PET/PVC multilayers” has been added for a brief description about the Favrini films in the revised manuscript.
- (3) In the line of 338, “see details in the Supplementary Information” has been corrected as “see Methods section for details”.

Reviewer #2 (Remarks to the Author):

In this manuscript, the authors have developed a novel Fabry-Perot nanocavity-type electrochromic device with rich and subtle color modulation, which is a breakthrough in electrochromic field, especially for inorganic materials. Based on the structural colors generated among layered W and WO_3 interfaces, an all-inorganic electrochromic device has been demonstrated with nearly full-color tunable range, providing a new strategy for the achievement of multichromatic electrochromism with inorganic materials. Since the design of device is conceptually novel together with interesting findings, this manuscript is suggested to be suitable for publishing in Nature Communications after minor revisions.

Comment 1:

In page 4 line 53, the authors claimed “Inorganic electrochromic materials lack the ability for subtle colour tuning”. This statement is not rigorous because V_2O_5 is one multicolor inorganic electrochromic material.

Author reply: Thank you for the valuable comments. The following statement has

been added.

“One particular example is vanadium oxide (V_2O_5), with the appearance in three colours (yellow \leftrightarrow green \leftrightarrow blue) during ion insertion, however, still plagued with problems of poor colour-tuning versatility, low colour saturation, and low coloration efficiency.^{19,20}”

Comment 2:

In page 4 and page 5, the authors described that the Figure 1 is the data of electrochromic device. However, it can be seen from Figure 1a, d and g, all the data should be the electrochromic electrode, not the electrochromic device.

Author reply: Thank you for the valuable comments. The relevant description of “electrochromic device” in Figure 1 has been replaced by the term “electrochromic electrode” in the revised manuscript, just as suggested.

Comment 3:

In Figure 2f, the reflectance valley of the F-P nanocavity-type electrode is observed to be red-shifted as the thickness of WO_3 increased. Could the authors explain for it?

Author reply: Thank you for the valuable comments. The reflectance valley of the F-P nanocavity-type electrode is red-shifted as the thickness of the WO_3 layer increased. These thickness-dependent shifts can be understood using the following equation:

$$\lambda_{DIV} = \frac{4n_1d}{2k + 1} \quad (k = 0, 1, 2 \dots)$$

where λ_{DIV} is the location of the wavelength intensity minima in the reflective spectrum (which has also been described as the destructive interference valley, DIV); d is the film thickness; and k is the DIV order. As can be clearly inferred from this equation, the wavelength positions of the DIV will be gradually red-shifted as the thickness of the tungsten oxide layer increases, thus resulting in different structural colors. Relevant information has been added in the revised manuscript.

Comment 4:

In Figure S8, the specific values of applied voltages and thicknesses of WO₃ should be provided.

Author reply: Thank you for the valuable comments. The specific values of applied voltages and the thicknesses of WO₃ have been provided in Supplementary Figure 8.

Comment 5:

I am curious weather the reflective metallic tungsten layer is unique in the Fabry-Perot nanocavity-type electrochromic device. Have the authors made a comparison between tungsten and other metals as reflective layers, for the performance and mechanism in the relative Fabry-Perot nanocavity-type electrochromic devices?

Author reply: Thank you for the valuable comments. According to common understanding, one of the greatest weaknesses of inorganic electrochromic materials is their monotonous colour changes. Thus, the ultimate goal of full-colour tunability for future electrochromic technology has been difficult to achieve with devices based on these typical materials. Our present work aims to broaden the colour versatility of inorganic electrochromic materials by introducing ultracompact Fabry-Perot (F-P)

nanocavities into relative electrochromic devices. For a proposed F-P nanocavity-type electrochromic device, the metal layer is the central component, as this layer acts as a reflecting mirror for the optical interference that allows the generation of different colours. Generally, the choice of metal layer in our work is based on the following criteria.

- (1) **Large dip depth.** As is well known, the depth of the resonance dip (that is, the difference in intensity between the dip minimum and the threshold) in a reflectance spectrum is an important indicator of resonance efficiency. The greater the depth of this dip, the better the efficiency of resonance that will be achieved. Supplementary Figure 17 shows the dependence of the depth of resonance dip on refractive index (n) and extinction factor (k) for the metal layers analysed in our work (for these measurements, 200-nm-thick WO_3 is set as the dielectric layer). The yellow region of the spectrum corresponds to a large dip depth, while the blue region indicates a small dip depth. As can be seen, the perceived (n , k) values for Al, Ag, Au and Cu are either fully or partly located in the blue region, indicating that the colour gamut was limited because a strong resonance could not be excited over the whole visible range. In contrast, the (n , k) values for Ni, Ti, Cr, V, Zn and W are all located in the yellow region, suggesting that a wide colour gamut is established by strong resonance behaviours operating across the entire visible spectrum.
- (2) **Good stability.** We choose metal layers with good environmental and electrochemical stability.
- (3) **Strong adhesion characteristics.** We choose metal layers that adhere well to the substrate and the electrochromic layer.

According to the above criteria, W is selected as the ideal choice for the proposed F-P nanocavity-type electrochromic device.

Supplementary Figure 17. The dependence of the depth of resonance dip on the refractive index (n) and extinction factor (k) of metal layers with different thicknesses. The parameters involved in the simulation are wavelength (400~800 nm) and WO_3 thicknesses (200 nm). The optical constant (n, k) of metal W is obtained by fitting the ellipsometry spectra using a quasi-static approach (Supplementary Figure 1, 2). The optical constants (n, k) of the other metals are all cited from the following website: <http://refractiveindex.info> (this source does not report an optical constant for metal W in the visible wavelength region).

Comment 6:

The references in this manuscript are few. There are some latest works focus on combination of structural and electrochromic effects (Angew. Chem. Int. Ed. 2016, 55, 2503–2506); amorphous WO_3 with ultra-large optical modulation (Chem. Sci., 2016, 7, 1373–1382); WO_3 /metal tungsten electrochromic electrode (Solar Energy Mater. Solar Cells, 2011, 95, 2107); novel chromic mechanism based on light scattering theory (ACS Appl. Mater. Interfaces 2018, 10, 37685); which are highly related to the present work, recommended to the authors for discussion.

Author reply: Thank you for the valuable comments. All of the references have been cited and discussed in the revised manuscript.

“So far, a few attempts have been made to construct structural colour-enhanced

electrochromic devices, including the introduction of opal/inverse opal photonic crystals,²²⁻²⁴ Bragg mirrors^{25,26} and Mie scattering²⁷ into electrochromic devices. These modifications have attempted to introduce the capacity to display additional colours by manipulating the photonic bandgaps of the periodic structures of electrochromic materials. However, to the best of our knowledge, rich and subtle colour adjustment has rarely been achieved by inorganic electrochromic materials incorporating these modifications, and this limitation has already become a bottleneck for the further development of electrochromic technology. Given this bottleneck, it is crucial to develop novel structures for inorganic electrochromic devices to broaden their colour palettes.”

Reviewer #3 (Remarks to the Author):

The authors present an interference-based approach to obtain distinct colors that may be further transitioned between a palette of available colors based on the electrochromic characteristics of the materials used. More precisely, color tuning is achieved by means of thickness control, while further color transitions are obtained via electrochemical control. Tungsten oxide and metallic tungsten are used as the base materials to obtain the interference patterns, in the architecture that authors describe as Ultracompact Fabry-Perot nanocavities. Overall, the manuscript presents a sound study that may impact the electrochromic field, opening new directions for broader color tunability in inorganic materials. Acceptance for publication in Nature Communications may be considered, providing several changes are properly addressed. Please let me suggest some of these changes/comments (not in order of importance, just randomly listed as they appear while reading the text):

Comment 1:

Please avoid the use of subjective adjectives or descriptions like “dexterously” (line 24 and 66) or “gorgeous” (line 169), in favor of a more aseptic and descriptive style.

Author reply: Thank you for the valuable comments. The words “dexterously” and

“gorgeous” have been removed in the revised manuscript.

Comment 2:

Lines 27-32 are directly copied from the main text (lines 71-75). Please elaborate a different sentence.

Author reply: Thank you for the valuable comments. To avoid duplication, the sentence in lines 71-75 has been modified in the revised manuscript.

“Accordingly, for this type of well-designed electrochromic device, a wide range of brilliant and highly saturated structural colours can be generated prior to applying voltages, with even more subtle chromatic states being achieved during the electrochromic process. Exceptionally, a rich variety of blue shades, such as sky blue, aqua blue, ocean blue, turquoise blue, peacock blue and navy blue, can be obtained as a result of the special structural design.”

Comment 3:

A more detailed description related to the results shown in figure 1 is missing. There is no information on how the simulated data were obtained. Please include a detailed explanation (whether in main text or Supp. Info).

Author reply: Thank you for the valuable comments. The description of the simulation methods has been provided, together with the models necessary for the simulation in the supporting information in the revised manuscript.

“Computational modelling of transmittance/reflectance spectra.

We use the characteristic matrix method derived from the basic principles of Maxwell’s equations to model the transmittance/reflectance spectra of the F-P nanocavities. MATLAB is used to edit the calculation code. A detailed description of this process is provided by the following formula:¹

$$\begin{bmatrix} B \\ C \end{bmatrix} = \begin{bmatrix} \cos \delta_1 & (i \sin \delta_1)/Y_1 \\ i \sin \delta_1 Y_1 & \cos \delta_1 \end{bmatrix} \begin{bmatrix} 1 \\ Y_2 \end{bmatrix}, \text{ (S1)}$$

where B and C is the normalised electric and magnetic fields at the front interface and δ_l is the phase thickness. This last property is defined as:

$$\delta_1 = \frac{2\pi N_1 d_1 \cos \theta_1}{\lambda}, \quad (\text{S2})$$

where N_1 and d_1 is the refractive index and the thickness of the first layer, respectively; θ_1 is the angle obtained from Snell's law (light is considered to have normal incidence for our study); and λ is the wavelength. Optical admittance values Y_1 and Y_2 are given by:

$$Y_1 = Y_0 N_1, \quad (\text{S3})$$

$$Y_2 = Y_0 N_2, \quad (\text{S4})$$

where Y_0 is the optical admittance in free space. The electric and magnetic components of Y_1 and Y_2 are equal.

The transmittance/reflectance spectra of the F-P nanocavities are thus calculated by equations (S5) and (S6), as shown below:

$$R = \left(\frac{Y_0 B - C}{Y_0 B + C} \right) \left(\frac{Y_0 B - C}{Y_0 B + C} \right)^*, \quad (\text{S5})$$

$$T = \frac{4N_2 Y_0^2}{(Y_0 B + C)(Y_0 B + C)^*}, \quad (\text{S6})$$

where (.)^{*} indicates a complex conjugate.”

Comment 4:

While reading the manuscript, one miss additional efforts on trying to teach the working principles and fundamentals of this architecture. Specially, it would be important to teach why color tuning works with one material as a reflector (W) but not with others (Au). Although it may be obvious for some specialists in the field, it may not be such for all people in the electrochromics community. This would also be particularly important and encouraged considering the broad and multidisciplinary audience of this journal.

Author reply: Thank you for the valuable comments. The key selection criteria of reflector layers in our architecture are summarized as follows:

“According to common understanding, one of the greatest weaknesses of inorganic electrochromic materials is their monotonous colour changes. Thus, the ultimate goal of full-colour tunability for future electrochromic technology has been difficult to achieve with devices based on these typical materials. Our present work aims to

broaden the colour versatility of inorganic electrochromic materials by introducing ultracompact Fabry-Perot (F-P) nanocavities into relative electrochromic devices. For a proposed F-P nanocavity-type electrochromic device, the metal layer is the central component, as this layer acts as a reflecting mirror for the optical interference that allows the generation of different colours. Generally, the choice of metal layer in our work is based on the following criteria.

- (1) **Large dip depth.** As is well known, the depth of the resonance dip (that is, the difference in intensity between the dip minimum and the threshold) in a reflectance spectrum is an important indicator of resonance efficiency. The greater the depth of this dip, the better the efficiency of resonance that will be achieved. Supplementary Figure 17 shows the dependence of the depth of resonance dip on refractive index (n) and extinction factor (k) for the metal layers analysed in our work (for these measurements, 200-nm-thick WO_3 is set as the dielectric layer). The yellow region of the spectrum corresponds to a large dip depth, while the blue region indicates a small dip depth. As can be seen, the perceived (n , k) values for Al, Ag, Au and Cu are either fully or partly located in the blue region, indicating that the colour gamut was limited because a strong resonance could not be excited over the whole visible range. In contrast, the (n , k) values for Ni, Ti, Cr, V, Zn and W are all located in the yellow region, suggesting that a wide colour gamut is established by strong resonance behaviours operating across the entire visible spectrum.
- (2) **Good stability.** We choose metal layers with good environmental and electrochemical stability.
- (3) **Strong adhesion characteristics.** We choose metal layers that adhere well to the substrate and the electrochromic layer.

According to the above criteria, W is selected as the ideal choice for the proposed F-P nanocavity-type electrochromic device.”

Relevant information has been added in the revised manuscript.

Comment 5:

As commented before for the simulations, information about the FDTD calculations should be included (in supp. Info). (lines 154-155) Related to those lines, there could be a mistake in the text. I don't see that the text corresponds to the refereed figures S1 and S2. Dependence of the resonance behavior vs thickness is not shown in those figures.

Author reply: Thank you for the valuable comments. The error has been corrected, as Supplementary Figure 1, 2 are not referred to on lines 154–155. The sentence on lines 154–155 is a leading sentence, which summarises the purpose of the paragraph. Furthermore, the detailed description of the FDTD simulation methods, together with the models necessary for the simulation, have been provided in the supporting information in the revised manuscript. Additionally, in the revised manuscript, the dependence of resonance behaviour on thickness is shown in Figure 3f, with a list of thickness values specified in the figure caption.

“Finite difference time domain (FDTD) simulations

The simulation model is illustrated in Supplementary Figure 3. Periodic boundary conditions along the x- and y-axes are implemented for the simulation in a unit cell of 400 nm. Perfectly matched layers are set according to the propagation of electromagnetic waves (z-axis). Planewave sources are launched incident to the unit cell along the backwards z-direction. Two time-monitors are added to the simulation mode, and reflectance spectra are collected with a reflectivity monitor placed behind the radiation source. The complex refractive indexes (optical constants) of WO₃ and metal W for the simulation are based on the data illustrated in Supplementary Figure 2.”

Supplementary Figure 3. FDTD simulation model for the F-P nanocavities on a substrate.

Comment 6:

Figure 2f. Please specify which lines (dashed or solid) correspond to simulated and measured data.

Author reply: Thank you for the valuable comments. The dashed lines correspond to simulated data, and solid lines correspond to measured data. We have specified them in the figure caption of Figure 2f in the revised manuscript.

Comment 7:

Lines 177- 182. I think refereed figures 4 and 5 are interchanged. Lines 178-180 correspond to fig 5 and the rest to fig 4?

Author reply: Thank you for the valuable comments. We have corrected for this in the revised manuscript.

Comment 8:

Line 184. Attending to the spectra shown, it would probably be better to state a $\pm 40^\circ$

insensitivity range, instead of $\pm 50^\circ$ as the authors claim. As a related comment, I would say that this angle range probably is not outstanding, compared with other image-forming competing technologies (electronic ink or LED). From my point of view, this is one of the weaknesses of this technology as far as it is presented now. The authors may want to comment on it and propose ways of overcoming or improving these limitations.

Author reply: Thank you for the valuable comments. The range of insensitivity has been changed from ± 50 to $\pm 40^\circ$ in the revised manuscript. Yes, we also believe that the angle-insensitive performance of our F-P nanocavity-type electrochromic devices needs to be further improved to expand their applications. A candidate strategy to overcome the challenge is to making noncontinuous nanocavities in our devices. Certainly, more efforts are required to confirm the idea in future studies.

Comment 9:

Line 193- The authors may want to find another equivalent expression to “rich-colours contained electrochromism”. I think is confusing.

Author reply: Thank you for the valuable comments. The expression has been changed to “significantly wide color gamut” in the revised manuscript.

Comment 10:

One of my main concerns about the presented technology is the lack of a transparent to colored transition. While multicolor transitions may be appealing, they are difficult to use in real devices. For instance, if the authors would like to develop a full color pixel based on a monolithic architecture, they will not have the possibility of showing a non- colored state, which greatly lowers the future development of this technology, and restricts it to a limited number of applications. In the same direction, the intrinsic architecture of these devices is limited to a reflective type. This compares unfavorably with other existing electrochromic approaches in which the reflective-transmissive behavior is translated to the electrolyte. The authors may have these aspects in mind in order to try to make a broader discussion on the future of this technology.

Author reply: Thank you for the valuable comments. One of the greatest challenges in the design of F-P nanocavity-type electrochromic devices is to find suitable ways to achieve reversible transitions between transparent and coloured states. Our preliminary study suggests that, significantly decreasing the thicknesses of metal layers in these devices to several nanometres may be a practical way to address this challenge, although subtle modification of other device parameters is needed. Further experimentation is still underway to verify whether our proposed strategy is indeed effective. Furthermore, we believe that electrochromic devices with these kinds of coloured-to-coloured transitions may find applications in interesting areas, such as “chameleon” cars.

Comment 11:

Line 295- Differences in the switching speeds are not so obvious as the authors claim. One could expect that ITO and FTO based devices would not show many differences as the conductivity values on these substrates is usually similar. Again, one would expect that a metallic substrate would enhance considerably the switching speed. It would be interesting to add conductivity values of each substrate. Also, surface values of the studied devices will help on this. For small surfaces, the effect of the conductivity of the substrate would be less critical.

Author reply: Thank you for the valuable comments.

A table summarizing the information about the electrical conductivity, substrate surface area, and switching time of the three substrates (ITO-based substrate, FTO-based substrate and metal W-based substrate) has been given as Supplementary Supplementary Table 2 in the revised manuscript. As can be seen from the table, the electrical conductivity of the substrates had a substantial effect on the electrochromic performances of the devices (particularly on switching times). Thus, given the same substrate surface area, the switching speed of the metal W-based F-P nanocavity-type electrochromic electrode is faster than the ITO-based electrochromic electrode but comparable to the FTO-based electrochromic electrode.

Supplementary Table 2. Summary of the information about the electrical conductivity, substrate surface area, and switching time of different substrates.

	FTO	ITO	W (100 nm)	W (200 nm)	W (300 nm)
Substrate surface area (cm ²)	4.5	4.5	4.5	4.5	4.5
Sheet resistance (Ω/\square)	10	20	10	8	7
Switching times (t_b/t_c , s)	2.9/4.2	7.5/8/7	3.3/3.1	3.1/2.3	2.9/2.2

Comment 12:

Long term cyclability. I'd suggest the authors to perform a longer study. Just 150 cycles seems clearly insufficient considering the conventional tests on the field.

Author reply: Thank you for the valuable comment. As shown in Supplementary Figure 14 in the revised manuscript, the stability assessment has been extended to 1000 cycles, still showing excellent cycling stability.

Comment 13:

Line 340- The authors refer to a bleached state. Please be cautious while using that word (I'd suggest not to use it in this context). As I mentioned before, there is no such

a bleached state in this architecture (understood as a transparent state, as usually referred in the field). “Light colored” or equivalent wording may be used.

Author reply: Thank you for the valuable comments. The expression “bleached and coloured states” has been replaced with “different colour states” in the revised manuscript.

Comment 14:

Lines 360-361. I think the verb is missing in the sentence.

Author reply: Thank you for the valuable comments. The verb “are” has been added in the revised manuscript.

Comment 15:

Conclusions: Please be cautious with the use of full color tunability, as it is not the same to achieve a full color palette of different films that achieving full color transitions electrochromically.

Author reply: Thank you for the valuable comments. The expression “with a nearly-full palette of colours” has been revised as “with very wide colour gamut distribution” in the revised manuscript.

Comment 16:

Some discussion on previous literature is missing in the introduction. Some references that mention or try Fabry-Perot interference based architectures are: <https://doi.org/10.1016/j.tsf.2013.10.030> <https://doi.org/10.1080/14786437308227562> or <https://doi.org/10.1117/12.621397>

Author reply: Thank you for the valuable comments. A paragraph discussing the Fabry-Perot interference-based architectures has been added, and all of the above references has been cited in the revised manuscript.

“Electrochromism, which denotes a reversible change in the electronic structure and optical properties (transmittance, reflectance, or absorption) of certain materials caused by stimulation by current or potential, has attracted intense scientific and

technological interests since Deb's pioneering studies due to its potential applications in displays, smart windows, and energy conservation devices.¹⁻⁷”

“A Fabry-Perot cavity is an optical resonator typically made from two facing parallel mirrors in which a light field can be selectively enhanced through resonance. Being simple and compact, these cavities have been frequently used in the design of certain optical devices, including tuneable optical filters, modulators and pressure sensors.²⁸⁻³¹ Aside from these limited examples, Fabry-Perot type cavities have rarely been incorporated into electrochromic devices, especially when those devices require complex light modulating capabilities.^{32,33}”

REVIEWERS' COMMENTS:

Reviewer #1 (Remarks to the Author):

I am totally satisfied with the replies and the revised manuscript. This manuscript can be published in Nature Communications.

Reviewer #2 (Remarks to the Author):

The manuscript is well revised. I recommend publishing this manuscript.

Reviewer #3 (Remarks to the Author):

The authors have made a significant effort to address all referees' comments, modifying the manuscript accordingly and including new sections/paragraphs when necessary.
I consider the manuscript is now acceptable for publication in Nature Comm in its present form.